# How Have Physical Activity and Sedentary Behavior, Changed during the COVID-19 Pandemic? A Swedish Repeated Cross-Sectional Design Study

**DOI:** 10.3390/ijerph20043642

**Published:** 2023-02-18

**Authors:** Daniel Lindberg, Maria Elvén, Kent W. Nilsson, Petra Von Heideken Wågert, Jonas Stier, Micael Dahlen, Birgitta Kerstis

**Affiliations:** 1Division of Social Work, School of Health, Care and Social Welfare, Mälardalen University, SE-72134 Västerås, Sweden; 2Division of Physiotherapy, School of Health, Care and Social Welfare, Mälardalen University, SE-72134 Västerås, Sweden; 3Center for Clinical Research, Central Hospital of Västerås, Uppsala University, SE-75236 Uppsala, Sweden; 4Division of Public Health Sciences, School of Health, Care and Social Welfare, Mälardalen University, SE-72134 Västerås, Sweden; 5Department of Marketing and Strategy, Stockholm School of Economics, SE-11383 Stockholm, Sweden; 6Division of Caring Sciences, School of Health, Care and Social Welfare, Mälardalen University, SE-72134 Västerås, Sweden

**Keywords:** COVID-19, life satisfaction, physical activity, sedentary behavior

## Abstract

Physical activity (PA) and sedentary behavior (SB) affect people’s physical and mental health. The aim was to examine changes in PA and SB in a Swedish population: at three time points: 2019, 2020, and 2022, i.e., before and during the COVID-19 pandemic. Pre-pandemic PA and SB, i.e., 2019, were assessed retrospectively in 2020. Associations between PA and SB with sex, age, occupation, COVID-19 history, weight change, health, and life satisfaction were also examined. The design was repeated cross-sectionally. The main findings demonstrate the PA levels decreased between 2019 and 2020, and between 2019 and 2022, but not between 2020 and 2022. The SB increase was most evident between 2019 and 2020. Between 2020 and 2022, results showed a decrease in SB, but SB did not reach pre-pandemic levels. Both sexes decreased their PA over time. Although men reported more PA sex, they did not have any association with PA changes. Two age groups, 19–29 years and 65–79 years, decreased their PA over time. Both PA and SB were associated with COVID-19, occupation, age, life satisfaction, health, and weight change. This study underlines the importance of monitoring changes in PA and SB as they have relevance for health and well-being. There is a risk that the levels of PA and SB do not return to pre-pandemic levels in the population.

## 1. Introduction

The worldwide coronavirus disease 2019 (COVID-19) has significantly changed people’s everyday lives, urging them to find new ways of coping based on the quality and resilience of their social and psychological framework [1]. According to the World Health Organization (WHO), COVID-19 has multiple consequences for health and well-being [2], including serious and life-threatening medical consequences, increased fatality and critical illness, and economic, social, and psychological consequences [3]. This study focuses on changes in physical activity (PA) and sedentary behavior (SB) during the COVID-19 pandemic.

The WHO describes PA as bodily movement produced by skeletal muscles that require energy expenditure during leisure time, transport, and work, and SB as sitting or lying down, including time spent sitting at work, watching television, and for other leisure activities [4]. PA and SB have wide-ranging effects on people’s physical and mental health [5,6,7,8,9]. PA can protect against negative health effects such as metabolic conditions, depression, and cognitive decline [10]. SB is associated with cardiovascular diseases, cancer, hypertension, depression, and metabolic disorders, such as diabetes mellitus and all-cause mortality [11]. Further, Eklund et al. (2021) describe that a decrease in SB is natural for people in the transition from working life to retirement but can be postponed by PA [12].

In the acute phase, the COVID-19 pandemic led to decreased PA and increased SB [13]. A Swedish study describes a greater decrease in PA among the oldest age group compared with the youngest [14]. Another Swedish study describes small changes during 2020, with negative changes in lifestyle habits and more time spent in a mentally passive state sitting at home, which is associated with higher odds of mental ill-health [15]. For many people, the pandemic restrictions have limited opportunities for PA and elevated SB [16]. Eventual negative consequences concerning decreased PA and increased SB may appear long after the COVID-19 pandemic. This study focused on the changes in levels of PA and SB in the Swedish population compared before and in two stages of the pandemic in December 2020, and January 2022. The Swedish pandemic strategy differs from many countries’ more restrictive measures [17] and favors voluntary compliance with recommendations [18].

Insufficient PA, tobacco use, and unhealthy diets are the most common causes of chronic diseases [19]. The restrictions to combat the spread of COVID-19 decreased gym access, outdoor PA, and increased dependence on at-home PA [20], which is counterproductive to health and well-being. Furthermore, PA reduces the impact of COVID-19, as it increases resistance to infectious diseases and mortality, as well as strengthens the outcomes of vaccination [21].

During the pandemic, vigorous PA decreased by 17% and walking time by 58% compared with before the pandemic [22]; SB increased, as well as overweight and obesity [23,24]. Low levels of education, unemployment, low income, being male, old age, and living in a socially vulnerable area increased the risks of serious consequences from COVID-19 [25]. Social and demographic factors such as education level, occupation, household income level, sex, age, and social vulnerabilities affect the levels of PA [25]. Concerning SB, short breaks and engaging in PA intermittently can have health benefits [26]. White-collar workers are at risk of developing SB associated with all-cause mortality [27]. It was therefore important to investigate factors associated with changes in PA and SB during the pandemic.

An association has been found between decreased PA and increased SB during the COVID-19 pandemic, especially among young people, students, and highly physically active men, measured before and during the pandemic in 2020 [22]. A recent study described that COVID-19 infection in unvaccinated athletes decreases respiratory function despite only reporting mild and few symptoms [28]. In conclusion, the PA level was affected not only in the general population but also in the athletic population with COVID-19 infection. Further, higher levels of PA are associated with greater psychological well-being [29]. In the early phase of the pandemic, mental health problems increased, especially in women and young adults [30]. A longitudinal study reported decreased step counts and increases in overeating and weight during the pandemic measured from February to June 2020 [31]. Therefore, it was important to investigate if occupation, life satisfaction, COVID-19 history, and weight change were associated with changes in PA and SB in a Swedish sample. Such an approach can pave the way for the identification of groups in need of specific health measures during and after a pandemic.

In a recent study by Elvén et al. (2022), changes in PA levels, types of PA, and sedentary behavior in the Swedish population before and during the COVID-19 pandemic were examined. Elvén’s study described that the majority had decreased PA levels with a concurrent increase in sedentary behavior during the COVID-19 pandemic.

This repeated cross-sectional design study is a follow-up study where Elvén’s data have been supplemented with another measurement in 2022 using the same instrument [32]. The study aimed to examine the changes in PA and SB levels in a Swedish population over time at three time points: before the COVID-19 pandemic in 2019, and during the pandemic in 2020 and 2022. This study thus examines whether the observed reduction in PA and increase in SB described in Elvén’s study (2022) remained or changed one year later in a Swedish population. Furthermore, associations between PA and SB with sex, age, occupation, life satisfaction, COVID-19 history, health, and weight change were investigated. To our knowledge, repeated measurements of PA and SB levels during COVID-19 have not been carried out previously.

## 2. Materials and Methods

### 2.1. Recruitment of Participants and Data Collection

This repeated cross-sectional population-based study included self-reported questionnaires for participants aged 18–79 years, recruited through a survey management service, including 50,000 potential participants. It was a population-based stratified sample, based on sex, age, and region in Sweden. The first survey was emailed to 2000 people on 7 December 2020. A follow-up email was sent on 11 December. The survey was anonymously completed by 1035 participants (52%). In addition, in the 2020 survey, the participants were asked to retrospectively self-report the same items about PA and SB during a normal week in December 2019, which was one year before the current time as well as three months before the WHO declared a worldwide COVID-19 pandemic. Data from 2019 and 2020 have been published by this research group, and this is a follow-up study to that article [32].

The second survey was emailed to 1894 new participants on 21 January 2022, with a follow-up email on 30 January, and was answered by 1095 participants (55%). All participants provided their informed consent to take part in the survey by answering the web-based questionnaire. The study was conducted according to the Helsinki Declaration [33] and Swedish law [34]. All personal data connections were deleted after the material was collected and were not accessible to the researchers in the present study.

### 2.2. Measures

The surveys included questions on demographic data, self-reported PA and SB during the last seven days, life satisfaction, COVID-19 history, and weight change. The International Physical Activity Questionnaire Short Form (IPAQ-SF) was used to self-estimate PA for the last 7 days [35]. Life satisfaction was measured with the question “How satisfied are you with life at the moment?” and “How would you rate your well-being right now, compared with 1 year ago?” on a ten-point Likert scale; a higher score indicated high life satisfaction. There was a question about confirmed COVID-19 (yes/no). The 2022 survey also included the question “Have you maintained, decreased, or increased your weight compared with before the pandemic?”

### 2.3. Analyses

Data were analyzed with descriptive statistics, using frequencies and percentages for categorical variables and means and standard deviations (SD) for continuous variables. To investigate the changes in PA and SB, a *t*-test, ANOVA, Mann–Whitney *U* test, Kruskal–Wallis with Scheffe post hoc test, and Spearman’s rho were used to avoid scaling artifacts. The outliers in PA were adjusted by a maximum of 70 h per week. Line bars illustrated visual changes in PA and SB concerning sex and age. We tested the independent variables sex, age, occupation, life satisfaction, COVID-19 history, health, weight change, and time (2019, 2020, and 2022) and their interactions in relation to the dependent variables PA and SB level with univariate analysis of variance. The independent variables were chosen based on previous research [5,6,7,8,22,23,24,25]. In the interaction model, only significant interactions were reported. The hierarchy in the model was created in relation to how much power the interaction had. This means that interactions with the highest power were interpreted as having the greatest impact compared to interactions with lower power. The observed power in the analyses for total PA was above 0.503, and the observed power in the univariate analyses of SB was above 0.607. As the distributions of the dependent variable PA were skewed, interactions for nonparametric tests based on aligned ranks were applied [36]. All tests were two-tailed, and the statistical significance was set at *p* ≤ 0.05. All analyses were conducted using IBM SPSS Statistics (Version 28.0; IBM SPSS, Armonk, NY, USA).

## 3. Results

### 3.1. Description of the Sample

The 2020 participants’ mean age was 50.6 years (SD = 16.5), with 49.6% women. The 2022 participants’ mean age was 52.2 years (SD = 16.5), with 51.0% women. In the data, nonmanual workers were overrepresented, as were respondents born in Sweden compared with the overall population [37]. Life satisfaction increased from 6.32 in 2020 to 6.72 in 2022 (*p* = 0.044). The number of confirmed COVID-19 cases increased from 5.3% in 2020 to 22.3% in 2022. A total of 33.4% reported weight gain during 2022 compared with 2019 (Table 1).

### 3.2. PA Changes over Time

The PA decreased over time between 2019 and 2020 (*p* < 0.001) and between 2019 and 2022 (*p* < 0.001), but not significantly between 2020 and 2022 for men as well as women, where men at all time points reported a higher PA than women. Furthermore, women had a more pronounced decrease in PA compared with men between the years 2020 and 2022, but the decrease was not significant (Table 2 and Figure 1). Although sex was correlated with PA, sex was not associated with the change in PA.

Two age groups, 18–29 years and 65–79 years, reported a decrease in PA over time. The age groups 18–29 and 30–49 years increased their PA in 2022 to the same level as in 2019. Furthermore, the age group 65–79 years reported a more pronounced decrease in PA over time (Figure 2).

### 3.3. SB Changes over Time

The SB increased over time between 2019 and 2020 (*p* < 0.001) and decreased between 2020 and 2022 (*p* < 0.001). Although the SB increased between 2019 and 2022, the change was not significant. Hence, SB “recovered” from 2020 to 2022, but not to the level of 2019 (Table 2). For both sexes, SB increased in 2020 compared with 2019, and although SB decreased in 2022 compared with 2020, SB did not reach the level it was before the COVID-19 pandemic. Between the sexes, there was no difference in SB over time (Figure 3). Three age groups (18–29 years, 30–49 years, and 65–79 years) reported significant changes in SB over time, whereas the age group 18–29 years reported the biggest increase in SB over time. In 2019, the age group 18–29 years was to a greater extent sedentary compared with other age groups (Figure 4).

### 3.4. Interaction Effects on PA

There was a main interaction between PA and confirmed COVID-19, occupation, age, life satisfaction, health, and weight gain. Hence, 9.8% of the PA was explained by the interaction of these factors. Confirmed COVID-19 and unemployment explained most of the PA. Confirmed COVID-19 and unemployment lead to a decreased PA over time. Age and life satisfaction, age and health, confirmed COVID-19 and self-employment, as well as blue-collar worker, life satisfaction and weight gain, and finally, life satisfaction and unemployment were also associated with PA (Table 3).

There was a negative correlation between decreased life satisfaction and weight gain over time (Spearman’s rho −0.219, *p* < 0.001). Those who reported in 2022 that they gained weight during the pandemic also reported lower levels of life satisfaction.

### 3.5. Interaction Effects on SB

There was a main interaction between SB and occupation, age, life satisfaction, health, and confirmed COVID-19. Furthermore, 19.1% of SB can be explained by the interaction of these factors. Being unemployed explained most of the interaction, followed by age and unemployment, life satisfaction and retirement, health and unemployment, age, being confirmed COVID-19 and self-employment, and finally health and self-employment (Table 4).

There was also a negative association between PA and SB in 2022 (Spearman’s rho −0.305 *p* < 0.001), indicating that those who decreased PA also increased SB during the pandemic.

## 4. Discussion

The aim was to examine whether the observed reduction in PA and increase in SB described in Elvén’s study (2022) had remained or changed one year later in a Swedish population [32]. Furthermore, associations between PA and SB with sex, age, occupation, life satisfaction, COVID-19 history, health, and weight change were investigated.

The main findings demonstrate the PA levels decreased between 2019 and 2020, and between 2019 and 2022, but not between 2020 and 2022. The SB increase was most evident between 2019 and 2020. Between 2020 and 2022, results showed a decrease in SB, but SB did not reach pre-pandemic levels.

Although sex was correlated with PA, i.e., men reported more PA than women at all measurement points, sex was not associated with the change in PA. Two age groups decreased their PA over time: those between 18–29 years and 65–79 years. The reduction in PA over time was more pronounced in the oldest age group. We described similar results with prior studies, as several scoping reviews have indicated a significant decrease in PA during the COVID-19 pandemic. According to Stockwell et al. (2020), PA decreased in the initial phase of the pandemic [38]. Caputo and Reichert (2021) concluded that most of the decrease in PA levels was due to social distancing measures. This study is a follow-up of a previous Swedish study that stated a decrease in PA before and during the pandemic [32]. The present study indicates that this negative trend continued over time. However, the reduction in PA was not as steep between 2020 and 2022 as it was between 2019 and 2020. Results, however, indicated a decrease in PA in the age groups between 18–29 years and 65–79 over time. This result is consistent with a previous study describing that the total PA in the older population decreased by 35% after the COVID-19 outbreak [39]. Furthermore, decreased PA in the age group 65–79 years could be explained by the fact that this age group was identified as being at risk of the most severe consequences of catching the virus. According to Eek et al. (2021), the oldest group (70+ years) had the highest odds of decreased PA compared with other age groups [14]. Our results are in line with Eek et al.’s (2021) finding that the negative trend in PA continued during the later stages of the pandemic. According to Elvén et al. (2022), PA decreased during COVID-19 [32], and this trend continued in 2022, as reported in this follow-up study. Further, the interaction model showed a more detailed picture of the interplay between PA and occupation, age, life satisfaction, health, and COVID-19. The model indicates that confirmed COVID-19 and being unemployed explained most of the PA level, followed by age and level of life satisfaction. It also indicated the direction of the interplay between PA, life satisfaction, and health. Our findings are consistent with a previous study [24] on how social and demographic factors are relevant to the level of PA. The interaction model further showed that COVID-19 infection was a significant factor explaining PA levels. Hence, COVID-19 infection may have negatively influenced respiratory functions and exercise performance, explaining some of the decreases in PA during the COVID-19 pandemic. Previous research has described that COVID-19 infection decreases respiratory muscle strength and pulmonary function in unvaccinated athletes [28].

SB increased for both sexes and for age groups 19–29 years, 30–49 years, and 65–79 years. However, there is an association between aging and increased SB, which could be postponed by PA [12]. Our results are consistent with previous studies [19,20] describing a significant increase in SB during the pandemic, compared with before the pandemic. There was also a correlation between increased PA and decreased SB, as described in a recent review [38].

Further, the interaction model showed a more detailed picture of the interplay between SB and occupation, confirmed COVID-19, age, life satisfaction, health, and weight gain. SB was primarily affected by the interaction of unemployment, as SB increased for this group. This could be explained by the fact that the pandemic made it difficult to search for and attain employment, as described [40]. Hence, this study suggests that there was an interaction between SB and social and demographic factors, for instance, occupation and age. COVID-19 infection did not have a significant effect on SB as in the case of PA.

More than 33% of the sample reported weight gain in 2022 compared with 2019. There was also a negative correlation between life satisfaction and weight gain, indicating that those who gained weight during the pandemic also reported less life satisfaction compared to those who did not gain weight. At the same time, overall life satisfaction increased from 2020 to 2022. This may be explained by less extensive restrictions compared with other countries [18], and that vaccination had begun. One may also speculate whether people have become accustomed to seeking new solutions for PA during the pandemic, which has a positive impact on life satisfaction [8]. Those who had the opportunity to work from home might have been able to adapt their activities and, as described by Birimoglu Okuyan (2022), maintain good levels of physical and mental health [41].

### Limitations and Strengths

One limitation was that this study did not allow for measuring effects on an individual level. It also precludes any cross-sectional design assumptions about temporality or causality between COVID-19 and PA and SB. Another limitation was the 48% and 45% drop-out rate and overrepresentation of individuals with a university degree (53% vs. 55%) compared with the overall population (29%), which might affect the representativeness of the sample [37]. Another limitation of the present study is that it primarily relies on self-reports and that the first survey also included recall about the PA and SB from one year ago, as the unreliability of human memory is well known [42]. We also speculate that the time point for the measurements affected the results; for instance, the levels of Omicron, a milder variant of the COVID-19 virus, were high in Sweden in January 2022. It may have been that PA and SB would be different during the summer, as COVID-19 was not then so aggressive; however, in this study, there were three measurement points at the same time of the year.

Strengths include the repeated population-based sample and the relatively large sample, which includes participants from 18 to 79 years. This has generated new knowledge about the consequences for the Swedish adult population and their PA and SB during the pandemic. Another strength is the use of nonparametric methods to validate our findings from the parametric analyses. The complementary statistical approaches eliminated scaling artifacts, one of the most ubiquitous sources of artifacts in interaction research.

## 5. Conclusions

There was a decrease in PA during both 2020 and 2022 in the Swedish population, with a concurrent increase in SB, compared to before the pandemic in 2019. Although there is a positive trend between 2020 and 2022, the levels of PA and SB have not reached their pre-pandemic levels. Society should be aware of the decreased PA and increased SB during the COVID-19 pandemic, as it may result in wide-ranging effects on people’s physical and mental health. Furthermore, the knowledge from this study can be used to promote PA and SB interventions as a way of promoting life satisfaction and health. This study also gives some suggestions on which groups are at risk of negative consequences during a pandemic. There is a risk that the levels of PA and SB do not return to at least the levels that they were before the pandemic, and this can not only negatively affect individuals but also public health. We underline the importance of repeated monitoring of changes and interventions in PA and SB, as they may have relevance for health and well-being in general.

## Figures and Tables

**Figure 1 ijerph-20-03642-f001:**
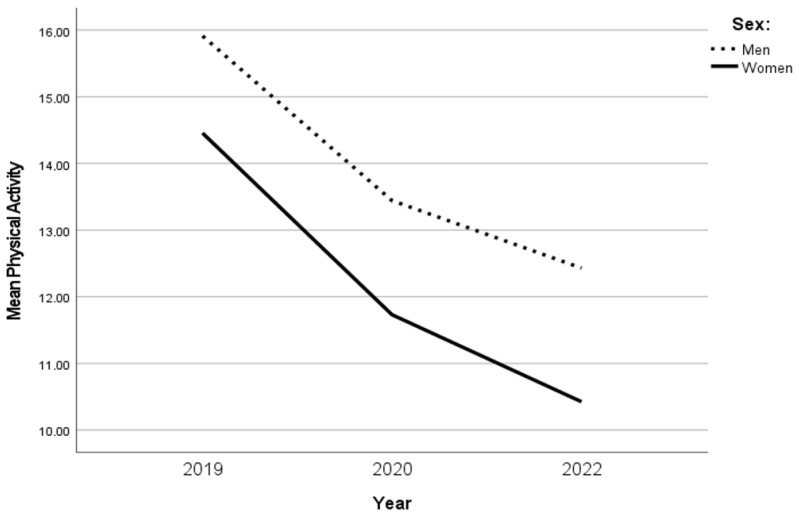
The changes in PA in sexes over time.

**Figure 2 ijerph-20-03642-f002:**
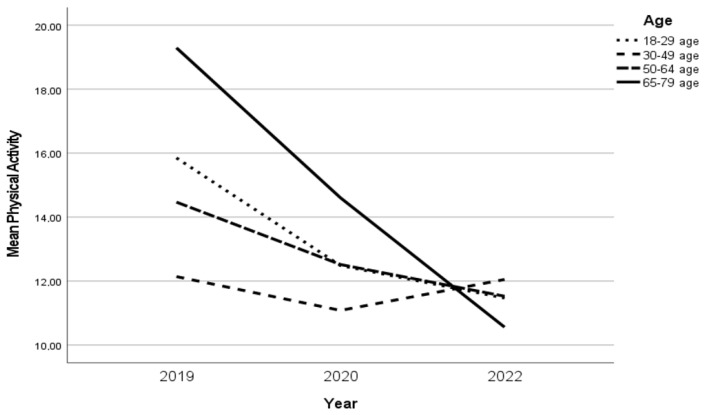
The changes in PA in age groups over time.

**Figure 3 ijerph-20-03642-f003:**
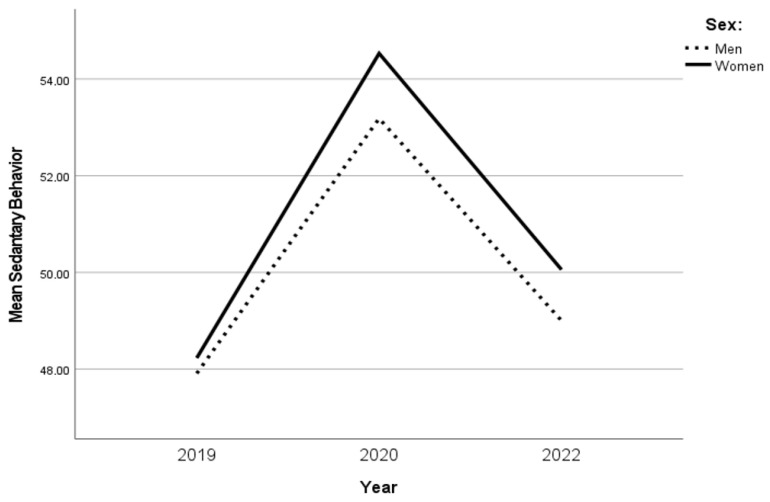
The changes in SB in sexes over time.

**Figure 4 ijerph-20-03642-f004:**
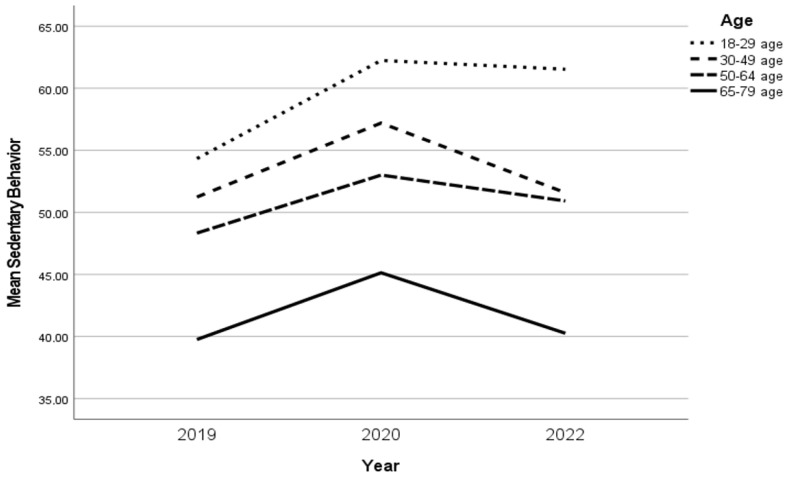
The changes in SB in the age groups over time.

**Table 1 ijerph-20-03642-t001:** Demographic characteristics for the two surveys.

Year	2020 *n* (%)	2022 *n* (%)
Highest education		
Compulsory school (9 years)	66 (6)	57 (6)
Senior high school	419 (41)	415 (39)
University	544 (53)	586 (55)
Other	2 (<1)	2 (<1)
Occupation		
Students	87 (8)	86 (8)
Manual workers	230 (22)	237 (22)
Nonmanual workers	334 (33)	334 (32)
Self-employed	62 (6)	53 (5)
Parental leave	12 (1)	15 (1)
Sick leave/early retired	25 (2)	28 (3)
Retired	241 (23)	283 (27)
Unemployed	30 (3)	20 (2)
Other	10 (1)	4 (<1)
Cohabitation status		
Not living with a partner	321 (48)	315 (30)31
Living with a partner	710 (62)	743 (70)
Origin		
Born in Sweden	926 (90)	984(93)
Mother born in Sweden	893 (87)	952 (90)
Father born in Sweden	850 (83)	895 (85)
Life satisfaction (mean rank *)	1.011	1.105
Confirmed COVID-19	55 (5.3)	244 (22.3)
Weight gain compared with before the pandemic	No data	366 (33.4)

* Mann–Whitney *U* test *p* < 0.001.

**Table 2 ijerph-20-03642-t002:** Physical activity (PA) and sedentary behavior (SB) over time.

	2019 & 2020 /2022 *n*	2019 PA Mean h/week	2020 PA Mean h/week	2022 PA Mean h/week	*p*-Value	2019 SB Mean h/week	2020 SB Mean h/week	2022 SB Mean h/week	*p*-Value
Total	1031/1060	15.2	12.6	11.5	**<0.001 ***	48.1	53.8	49.5	**<0.001 ****
Men	520/519	15.9	13.4	12.4	**0.002 ***	47.9	53.1	49.0	**0.002 ***
Women	511/541	14.5	11.7	10.4	**<0.001 ***	48.2	54.5	50.0	**<0.001 ****
18–29 years	171/125	15.9	12.5	11.5	**0.016 ***	54.3	62.2	61.5	**0.010 ****
30–49 years	317/314	12.1	11.1	12.1	0.858	51.2	57.2	51.6	**0.004 ****
50–64 years	288/289	14.5	12.5	11.5	0.121	48.3	53.0	50.9	0.077
**65–79 years**	259/294	19.3	14.6	10.6	**<0.001 ***	39.8	45.1	40.2	**0.007 ****

* Kruskal–Wallis test, ** ANOVA, **bold** = significant.

**Table 3 ijerph-20-03642-t003:** Interaction model for PA.

Variables	F	*p*-Value	Observed Power	Interaction Increase/Decrease
Unemployed * Confirmed COVID-19	16.4	<0.001	0.981	Low PA was associated with unemployment and confirmed COVID-19
Age * Life satisfaction	9.1	0.003	0.854	Low PA was associated with low age and low life satisfaction
Age * Health	8.5	0.004	0.827	Low PA was associated with high age and good health
Self-employed * Confirmed COVID-19	8.0	0.005	0.807	Low PA was associated with self-employment and confirmed COVID-19
Blue-collar worker * Confirmed COVID-19	6.5	0.011	0.719	Low PA was associated with being blue-collar worker and confirmed COVID-19
Life satisfaction * Weight gain	5.0	0.027	0.601	Low PA was associated with low life satisfaction and weight gain
Unemployed * Life satisfaction	4.0	0.047	0.510	Low PA was associated with unemployment and high life satisfaction

* = in combination with.

**Table 4 ijerph-20-03642-t004:** Interaction model for SB over time.

Variables	F	*p*-Value	Observed Power	Interaction Increase/Decrease
Unemployed	21.2	<0.001	0.996	High SB was associated with unemployment
Unemployed * Age	15.0	<0.001	0.971	Low SB was associated with unemployment and high age
Retired * Life satisfaction	14.7	<0.001	0.968	Low SB was associated with retirement and high life satisfaction
Unemployed * Health	12.0	<0.001	0.932	Low SB was associated with unemployment and good health
Age	10.7	0.001	0.902	Low SB was associated with high age
Self-employed * Confirmed COVID-19	8.0	0.005	0.803	High SB was associated with self-employment and confirmed COVID-19
Self-employed * Health	5.0	0.026	0.607	Low SB was associated with self-employment and good health

* = in combination with.

## Data Availability

The data presented in this study are openly available in FigShare at https://doi.org/10.6084/m9.figshare.21802941.v1.

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
