# Peer review of "How Have Physical Activity and Sedentary Behavior, Changed during the COVID-19 Pandemic? A Swedish Repeated Cross-Sectional Design Study"

_ijerph, 2023, doi:10.3390/ijerph20043642_

Round 1
Reviewer 1 Report
Dear authors,
Thank you very much for the effort and effort you put forth in the research. Your research is valuable both in terms of its design and subject, and I thank you for going by creating titles that make it easier to read during the writing phase. After a few minor verifications below, I think your article is suitable for publication in ijerph.
Introduction
-The introduction is very descriptive and written in detail on the name of PA, but I recommend you to add it by referring to the study I will give below, which proves that PA levels can be affected depending on outputs such as respiratory functions that affect PA levels, not just PA. In addition, the PA level was affected not only in normal individuals but also in the athletic population with COVID.
Bostancı, Ö., Karaduman, E., Çolak, Y., Yılmaz, A. K., Kabadayı, M., & Bilgiç, S. (2023). Respiratory muscle strength and pulmonary function in unvaccinated athletes before and after COVID-19 infection: A prospective cohort study. Respiratory Physiology & Neurobiology, 308, 103983.
-I think that the Method and Result sections are sufficient.
-I would recommend adding a few additions to the Discussion section that talks about the effects of COVID-19 in the athletic population and draws conclusions.
Yours sincerely
Author Response
Dear review # 1
International Journal of Environmental Research and Public Health
Thank you for the opportunity to resubmit the revised manuscript ‘Changes in Physical Activity and Sedentary Behavior Before and During the COVID-19 Pandemic: A Swedish Repeated Cross-sectional Design Study’. Below are the authors’ responses to the reviewer’s comments. All revisions are marked with yellow in the manuscript, and each response is accompanied by a reference to the text with page and line numbers where appropriate.
Kind regards
Birgitta Kerstis
_______________________________________
The authors would like to thank reviewer 1 for insightful and adequate comments which raised the quality of the paper.
We thank reviewer 1 for the positive feedback.
_______________________________________________________
-The introduction is very descriptive and written in detail on the name of PA, but I recommend you add it by referring to the study I will give below, which proves that PA levels can be affected depending on outputs such as respiratory functions that affect PA levels, not just PA. In addition, the PA level was affected not only in normal individuals but also in the athletic population with COVID.
Response: We have added the suggested reference (p 2, lines 97-100).
_______________________________________________________
-I think that the Method and Result sections are sufficient.
Response: Thank you for this positive feedback.
_______________________________________________________
-I would recommend adding a few additions to the Discussion section that talks about the effects of COVID-19 in the athletic population and draws conclusions.
Response: The text has been revised with additions to the discussion section about the effects of COVID-19 in the athletic population (p 11, line 300-306).

Reviewer 2 Report
The study assessed the changes in physical activity (PA) and sedentary behavior (SB) during the COVID-19 pandemic by comparing cross-sectional population-based data collected in 2020 and 2022. The results show an overall decrease in PA and increase in SB during the pandemic compared with the pre-pandemic situation (assessed retrospectively in 2020), whereas between 2020 and 2022, there was no change in PA and SB decreased.
By providing population-based data from different stages of the COVID-19 pandemic the study has a potential to make a nice contribution to the literature about the effects of pandemic on health-related behaviors, however, the manuscript would benefit from a revision to improve its clarity.
Major points:
· As mentioned in methods section, this is a follow-up to a previous study by this research group (Elvén et al. Int J Environ Res Public Health 2022, 19(5)). Thus, it would be helpful to include a short overview of the main findings of the previous article in the introduction. Also, the authors could put more effort to show the value of the follow-up by clearly stating its rationale and how does it extend the previous study.
· I find the description of interaction analyses (sections 2.3, 3.4 and 3.5) confusing and difficult to follow. It is not clear to me what was done and how. E.g., “We tested the independent variables sex, age, occupation, life satisfaction, COVID-19 history, health, weight change, and time, and their interactions in relation to the dependent variables PA and SB with univariate analysis of variance” (lines 140-142). Considering that PA and SB were assessed at three time points, which PA and SB were treated as dependent variables? If data for all three time points were pooled, was the time included as an independent variable in the models? How were the independent variables chosen for the interaction analyses? Tables 3 and 4 seem to omit statistically insignificant interactions but this is not clearly stated in the text. How are the estimates of observed power interpreted?
Minor points:
· The first part of the title repeats word for word the title of the previous article. To avoid possible confusions in the future, I would suggest changing the title.
· Please make sure that all the authors are listed correctly. Micael Dahlen is listed twice and B. K. is not listed at all although their initials appear repeatedly in the section of author contributions.
· Please make it clear already in the abstract that pre-pandemic PA and SB were assessed retrospectively in 2020, otherwise it creates false expectations.
· There are contradictory statements about the association between PA and sex: „Both sexes decreased their PA over time, but sex did not have any association with PA“ (line 20), “… where men at all time points reported higher PA than women (lines 164-165), and “Sex did not correlate with PA although men reported more PA than women at all measure points” (lines 231-232). Based on my understanding, if men reported more PA than women, it means that sex was correlated with PA. Or was it meant that sex was not associated with the change in PA? Please clarify.
· Lines 165-166: “Furthermore, women had a more pronounced decrease in PA compared with men between the years 2020 and 2022 but were not significant”. The sentence is incomplete, it is not clear what was not significant.
Author Response
Dear Review # 2
International Journal of Environmental Research and Public Health
Thank you for the opportunity to resubmit the revised manuscript ‘Changes in Physical Activity and Sedentary Behavior Before and During the COVID-19 Pandemic: A Swedish Repeated Cross-sectional Design Study’. Below are the authors’ responses to the reviewer’s comments. All revisions are marked with yellow in the manuscript, and each response is accompanied by a reference to the text with page and line numbers where appropriate.
Kind regards
Birgitta Kerstis
The study assessed the changes in physical activity (PA) and sedentary behavior (SB) during the COVID-19 pandemic by comparing cross-sectional population-based data collected in 2020 and 2022. The results show an overall decrease in PA and increase in SB during the pandemic compared with the pre-pandemic situation (assessed retrospectively in 2020), whereas between 2020 and 2022, there was no change in PA and SB decreased.
By providing population-based data from different stages of the COVID-19 pandemic the study has a potential to make a nice contribution to the literature about the effects of pandemic on health-related behaviors, however, the manuscript would benefit from a revision to improve its clarity.
Response: Thank you for this positive feedback.
_______________________________________________________
- As mentioned in methods section, this is a follow-up to a previous study by this research group (Elvén et al. Int J Environ Res Public Health 2022, 19(5)). Thus, it would be helpful to include a short overview of the main findings of the previous article in the introduction.
Response: The text has been revised according to these comments and suggestions (p 3, lines 110-123).
Also, the authors could put more effort to show the value of the follow-up by clearly stating its rationale and how does it extend the previous study.
Response: We have more clearly stated the rationale (p 3, lines 124-125).
- I find the description of interaction analyses (sections 2.3, 3.4 and 3.5) confusing and difficult to follow. It is not clear to me what was done and how. E.g., “We tested the independent variables sex, age, occupation, life satisfaction, COVID-19 history, health, weight change, and time, and their interactions in relation to the dependent variables PA and SB with univariate analysis of variance” (lines 140-142). Considering that PA and SB were assessed at three time points, which PA and SB were treated as dependent variables? If data for all three time points were pooled, was the time included as an independent variable in the models? How were the independent variables chosen for the interaction analyses? Tables 3 and 4 seem to omit statistically insignificant interactions but this is not clearly stated in the text. How are the estimates of observed power interpreted?
Response: The text has been revised according to these comments and suggestions (p 4, lines 170-176).
_______________________________________________________
- The first part of the title repeats word for word the title of the previous article. To avoid possible confusions in the future, I would suggest changing the title.
Response: The title has been changed (p 1, lines 2-4).
_______________________________________________________
- Please make sure that all the authors are listed correctly. Micael Dahlen is listed twice and B. K. is not listed at all although their initials appear repeatedly in the section of author contributions.
Response: Sorry for this, we have now changed (p 1, lines 8-12).
_______________________________________________________
- Please make it clear already in the abstract that pre-pandemic PA and SB were assessed retrospectively in 2020, otherwise it creates false expectations.
Response: We agree and have clarified the abstract (p 1, lines 25--26).
_______________________________________________________
- There are contradictory statements about the association between PA and sex: Both sexes decreased their PA over time, but sex did not have any association with PA“ (line 20), “… where men at all time points reported higher PA than women (lines 164-165), and “Sex did not correlate with PA although men reported more PA than women at all measure points” (lines 231-232). Based on my understanding, if men reported more PA than women, it means that sex was correlated with PA. Or was it meant that sex was not associated with the change in PA? Please clarify.
Response: We agree and have clarified (p 1, lines 28-34 and p 5 lines 201-202).
_______________________________________________________
- Lines 165-166: “Furthermore, women had a more pronounced decrease in PA compared with men between the years 2020 and 2022 but were not significant”. The sentence is incomplete, it is not clear what was not significant.
Response: We agree and have clarified the sentences (p 5, lines 201-202).
_______________________________________

Reviewer 3 Report
This paper addresses an important area of public health by examining changes in physical activity and sedentary behaviour as a result of pandemic influences. The topic of the article fits perfectly into the special issue of the journal.
The methodology of the empirical research has been developed and clearly explained by the authors so that the general public can respond to the aims of the paper. The article contains an appropriate number, quality and relevance of references. The tables are easy to understand and provide good support for the results.
The results are well presented because the paper shows a good coherence between the introduction, the objectives of the methodology and the discussion. The conclusions are also clear.
The effect of COVID on PA and SB has been reported in several publications, and the great value of the study is that it was conducted with data from a country that used an alternative epidemiological treatment. However, the novelty of the study is somewhat questionable. The hypotheses tested are also too general and trivial. I suggest to further strengthen the discussion and conclusion chapters and to draw deeper and more forward-looking conclusions.
Other comments:
- In the first chapter it is not necessary to create a subsection of a few lines for the hypotheses.
- In the second chapter, there is no justification for bolding some words.
- In the first table, the use of the mean to measure "life satisfaction" is methodologically incorrect, as it is an ordinal Likert scale.
- In the second table, there is a typo in the first column where "30-49 years" is replaced by "39-49 years". In the same table, in the last row, one of the numerical values (third column) is sorted higher than the other values.
- Figures 1-4 show the same thing as the previous table, but it may not be justified to present the same results in a simple line graph.
- The ages in lines 171 and 172 are probably incorrect because the two sentences contradict each other.
- The vertical dimensions of tables 3 and 4 could be reduced (by resizing the columns), they are unnecessarily large.
Author Response
Dear Review # 3
International Journal of Environmental Research and Public Health
Thank you for the opportunity to resubmit the revised manuscript ‘Changes in Physical Activity and Sedentary Behavior Before and During the COVID-19 Pandemic: A Swedish Repeated Cross-sectional Design Study’. Below are the authors’ responses to the reviewer’s comments. All revisions are marked with yellow in the manuscript, and each response is accompanied by a reference to the text with page and line numbers where appropriate.
Kind regards
Birgitta Kerstis
The authors would like to thank reviewer 3 for insightful and adequate comments which raised the quality of the paper.
This paper addresses an important area of public health by examining changes in physical activity and sedentary behaviour as a result of pandemic influences. The topic of the article fits perfectly into the special issue of the journal.
The methodology of the empirical research has been developed and clearly explained by the authors so that the general public can respond to the aims of the paper. The article contains an appropriate number, quality and relevance of references. The tables are easy to understand and provide good support for the results.
The results are well presented because the paper shows a good coherence between the introduction, the objectives of the methodology and the discussion. The conclusions are also clear. The effect of COVID on PA and SB has been reported in several publications, and the great value of the study is that it was conducted with data from a country that used an alternative epidemiological treatment.
Response: We thank reviewer 3 for the positive feedback.
_______________________________________________________
However, the novelty of the study is somewhat questionable. The hypotheses tested are also too general and trivial.
Response: The novelty of the study has been developed as well as the hypotheses (p 3, lines 115-125).
_______________________________________________________
I suggest to further strengthen the discussion and conclusion chapters and to draw deeper and more forward-looking conclusions.
Response: We agree and have strengthened the discussion and conclusion chapters (p 11, lines 267-270, 301-305, 308-309, p 12 lines 356-359).
_______________________________________________________
- In the first chapter it is not necessary to create a subsection of a few lines for the hypotheses.
Response: We agree and have added a subsection of a few lines for the hypotheses (p 3, lines 115-125).
_______________________________________________________
- In the second chapter, there is no justification for bolding some words.
Response: We have now removed the bolding in the second chapter.
_______________________________________________________
- In the first table, the use of the mean to measure "life satisfaction" is methodologically incorrect, as it is an ordinal Likert scale.
Response: We are grateful for pointing this out and have now changed to medians instead (p 5, line 194).
_______________________________________________________
- In the second table, there is a typo in the first column where "30-49 years" is replaced by "39-49 years". In the same table, in the last row, one of the numerical values (third column) is sorted higher than the other values.
Response: We have now changed (p 6, line 206).
_______________________________________________________
- Figures 1-4 show the same thing as the previous table, but it may not be justified to present the same results in a simple line graph.
Response: We agree to a certain extent but think that the figures make the result even more manageable and would therefore prefer to keep them (p 7 and 8).
_______________________________________________________
- The ages in lines 171 and 172 are probably incorrect because the two sentences contradict each other.
Response: We are grateful for pointing this out and have now changed to clarify (p 4, lines 171-176).
_______________________________________________________
- The vertical dimensions of tables 3 and 4 could be reduced (by resizing the columns), they are unnecessarily large.
Response: We agree and have resized the columns in all tables.
_______________________________________________________

Round 2
Reviewer 2 Report
The authors have done a good job in revising the manuscript but there are still a couple of minor issues that need to be corrected.
Section 2.2 mentions the question “Did COVID-19 cause the change in PA and SB?” (line 153 in the revised manuscript), but I don’t see that this question was analyzed in this article.
Lines 218-219: “The SB increased over time between 2019 and 2020 (p < 0.001), and between 2020 and 2022 (p < 0.001), but not significantly between 2019 and 2022.” Based on Table 2, SB decreased between 2020 and 2022.
Author Response
Dear Review # 2
International Journal of Environmental Research and Public Health
Thank you for the opportunity to resubmit the revised manuscript ‘Changes in Physical Activity and Sedentary Behavior Before and During the COVID-19 Pandemic: A Swedish Repeated Cross-sectional Design Study’. Below are the authors’ responses to the reviewer’s comments. The new revisions are marked with tracked changes in the manuscript, and the old changes in yellow.
Comments and Suggestions for Authors
The authors have done a good job in revising the manuscript but there are still a couple of minor issues that need to be corrected.
Response: Thank you for this positive feedback.
_______________________________________________________
Section 2.2 mentions the question “Did COVID-19 cause the change in PA and SB?” (line 153 in the revised manuscript), but I don’t see that this question was analyzed in this article.
Response: Thank you for observing this, we have now removed this sentence as this question was not analysed in this article.
_______________________________________________________
Lines 218-219: “The SB increased over time between 2019 and 2020 (p < 0.001), and between 2020 and 2022 (p < 0.001), but not significantly between 2019 and 2022.” Based on Table 2, SB decreased between 2020 and 2022.
Response: Thank you for observing this, we have now changed the sentence to:
The SB increased over time between 2019 and 2020 (p < 0.001) and decreased between 2020 and 2022 (p < 0.001), although the SB increased between 2019 and 2022 the change was not significant.
_______________________________________________________